# All-People-Test-Based Methods for COVID-19 Infectious Disease Dynamics Simulation Model: Towards Citywide COVID Testing

**DOI:** 10.3390/ijerph191710959

**Published:** 2022-09-02

**Authors:** Xian-Xian Liu, Jie Yang, Simon Fong, Nilanjan Dey, Richard C. Millham, Jinan Fiaidhi

**Affiliations:** 1Department of Computer and Information Science, University of Macau, Taipa, Macau SAR 519000, China; 2Chongqing Industry & Trade Polytechnic, Chongqing 408000, China; 3Department of Computer Science and Engineering, JIS University, Kolkata 700109, India; 4ICT & Society Group, Durban University of Technology, Durban 4001, South Africa; 5e-Health Research Group, Computer Science Department, Lakehead University, Thunder Bay, ON P7B 5E1, Canada

**Keywords:** SETPG (A + I) RD + APT model, COVID-19 sampling pooling, mobile app precise tracking technology, transmission dynamics

## Abstract

The conversion rate between asymptomatic infections and reported/unreported symptomatic infections is a very sensitive parameter for model variables that spread COVID-19. This is important information for follow-up use in screening, prediction, prognostics, contact tracing, and drug development for the COVID-19 pandemic. The model described here suggests that there may not be enough researchers to solve all of these problems thoroughly and effectively, and it requires careful selection of what we are doing and rapid sharing of results and models and optimizing modeling simulations with value to reduce the impact of COVID-19. Exploring simulation modeling will help decision makers make the most informed decisions. In order to fight against the “Delta” virus, the establishment of a line of defense through all-people testing (APT) is not only an effective method summarized from past experience but also one of the best means to effectively cut the chain of epidemic transmission. The effect of large-scale testing has been fully verified in the international community. We developed a practical dynamic infectious disease model-SETPG (A + I) RD + APT by considering the effects of the all-people test (APT). The model is useful for studying effects of screening measures and providing a more realistic modelling with all-people-test strategies, which require everybody in a population to be tested for infection. In prior work, a total of 370 epidemic cases were collected. We collected three kinds of known cases: the cumulative number of daily incidences, daily cumulative recovery, and daily cumulative deaths in Hong Kong and the United States between 22 January 2020 and 13 November 2020 were simulated. In two essential strategies of the integrated SETPG (A + I) RD + APT model, comparing the cumulative number of screenings in derivative experiments based on daily detection capability and tracking system application rate, we evaluated the performance of the timespan required for the basic regeneration number (*R*0) and real-time regeneration number (*R*0*t*) to reach 1; the optimal policy of each experiment is available, and the screening effect is evaluated by screening performance indicators. with the binary encoding screening method, the number of screenings for the target population is 8667 in HK and 1,803,400 in the U.S., including 6067 asymptomatic cases in HK and 1,262,380 in the U.S. as well as 2599 cases of mild symptoms in HK and 541,020 in the U.S.; there were also 8.25 days of screening timespan in HK and 9.25 days of screening timespan required in the U.S. and a daily detectability of 625,000 cases in HK and 6,050,000 cases in the U.S. Using precise tracking technology, number of screenings for the target population is 6060 cases in HK and 1,766,420 cases in the U.S., including 4242 asymptomatic cases in HK and 1,236,494 cases in the U.S. as well as 1818 cases of mild symptoms in HK and 529,926 cases in the U.S. Total screening timespan (TS) is 8.25~9.25 days. According to the proposed infectious dynamics model that adapts to the all-people test, all of the epidemic cases were reported for fitting, and the result seemed more reasonable, and epidemic prediction became more accurate. It adapted to densely populated metropolises for APT on prevention.

## 1. Introduction

The spread of novel coronavirus pneumonia (NCP) is more contagious than SARS. Although its mortality rate at 2.3% is lower than SARS at 9.6%, the damage that comes along with the travel quarantine and lockdown are destructive to the world economy. From bad to worse, by the mutations of coronavirus, many patients with COVID-19 become asymptomatic or show very mild symptoms. They can bypass basic temperature checks and are contagious enough to transmit the virus to others [1]. The epidemic currently has rapidly spread to many countries in the world. There is still a lack of scientific explanation for (1) super spreaders; (2) the precise incubation period; and (3) characterizing a simulation model with complex epidemic factors in multiple compartments.

In theoretical research, mathematical models play an extremely important role. They clearly reveal the main characteristics of infectious diseases through hypotheses, parameters, variables, and connections between populations. The results of the mathematical model can provide many powerful theoretical foundations and concepts. It has become a consensus to use mathematical models to help discover the spreading mechanism of infectious diseases and predict the epidemic trend of infectious diseases. It is of great practical significance to use nonlinear dynamics to establish mathematical models of infectious diseases to study whether infectious diseases will spread, continue, and whether they will eventually be eradicated [2] in a city. It can predict the trend of infectious diseases. It also provides useful information and effective measures for people to prevent and treat infectious diseases. The work reported in this paper contributes to analyzing and predicting the development of the epidemic through a compartmental simulation model embracing the nuclei acid screening factor for all residents. It shall be useful as a reference for the government’s epidemic-prevention decision-making policy.

The reporting of the local spread of the epidemic in Guangzhou began on 21 May 2021. Guangdong has discovered multiple virus mutants imported from abroad, involving countries such as the United Kingdom, South Africa, Nigeria, Brazil, and India. The Shenzhen outbreak in June 2021 was caused by the mutant strain that was first discovered in the United Kingdom. Recently, as the impact of the spillover of the epidemic in Brazil, India, and Peru continues to appear, and the number of newly confirmed cases in a single day in many Southeast Asian countries has rebounded. The hope for endemic is not optimistic. India, currently, as of July 2021, has the second-largest number of people infected with the COVID-19 in the world, and the epidemic is still developing at an alarming rate. As of 30 May 2021, there have been 23 confirmed cases and 7 asymptomatic infections in Guangzhou. The traceability of the virus showed that the genetic sequencing results of infected persons in the Liwan District of Guangzhou are highly homologous, and they are all the variants of the new coronavirus (B.1.617) previously discovered in India that spread extremely quickly. In order to quickly cut off the virus transmission route, Guangzhou decided to further expand the scope of nucleic acid testing on the basis of continuing nucleic acid screening in Liwan District from 30 May 2021 [3].

Virus mutation is a key variable affecting the prevention and control of the current global epidemic [4]. In the face of constantly mutating viruses, it is important to increase the screening rate of the entire population as soon as possible while increasing the vaccination rate and speeding up the construction of the domestic herd immunity barrier [5]. The global epidemic still spread at a high rate, and the pressure on imported cases prevention is still huge. When the immune barrier is constructed [6], even in the face of stronger mutant viruses, a good response ability in order to prevail in the race against the virus is very much needed.

The Hong Kong (HK) government is one of the pioneers after mainland China implemented the all-people-test (APT) program on 1 September 2020. Therefore, was taken as an APT case study for us to develop a pandemic prediction model. In the first week, about 310,000 people had registered. Under the epidemic, HK has been locked down for more than half a year; and the order of the social gathering ban has greatly affected the tourism, catering, and retail industries. As a strategy, nucleic acid testing is by far one of the most effective methods to sort out the infected from the population, preventing further viral infection. This strategy may work well for densely populated cities or areas, such as Hong Kong. APT could be relatively easier to implement, as it makes little difference between selectively testing only certain clusters and the whole city. As many places began to carry out large-scale nucleic acid screening, a large number of samples is required. Sample pooling is a quick approach to mass test many people by grouping several samples into one test. Sample pooling speeds up the whole disease screening process. When conducting large-scale population testing, typically 5–10 samples can be mixed for preliminary screening to improve testing efficiency and reduce testing costs [7,8]. Sample pooling is a solution to the testing pressure of large samples [9]. After applying some combinatorial optimization, the impact of technical defects on the results can be avoided as much as possible.

**Problem statements.** Given the unprecedented, fast-moving health and economic impacts of COVID-19, with citywide COVID testing becoming the norm, as planned by the government, a more dynamic forecasting approach was needed to leverage fast-changing external data and adaptive predictive models to inform an epidemic outlook. We need a strong model that predicts how the virus could spread across different countries and regions. The goal of this task is to build a model that predicts the spread of the virus and when it will end under different citywide COVID testing strategies and generate a best solution from the potential solutions that harvested daily external information on local virus and social policy impacts as well as data related to the impact of multiple groups and parameters on the outbreak, taking the latest data from countries on impact and correlating them with relevant policies.

**Highlights.** The innovative contributions of the presented study for Citywide COVID Testing (CTT) for COVID-19 models are highlighted as follows:By establishing a micro-epidemic prevention and control mechanism, the SETPG (A + I) RD + APT model takes into account a more complex population network, adding several key features of asymptomatic and symptomatic carrier transmission, especially for individuals with mild infection, to help scientific researchers develop insights that may contribute to public health policymaking, including contributions to public health-forecasting teams.A more realistic reconstruction of the pandemic situation would be to take into account epidemic prevention policies in the model, when virus carriers are found or when the number of infected exceeds a threshold determined by the capacity of the regional health care system, including the specific implementation of containment and putting other social distancing measures, such as “intermittent lockdown”, in place.Moreover, attention must be paid to the potential risk posed by re-infection, which is especially of concern with new variants.This model can also inform resource requirements of citywide COVID testing diagnostic capacity and the changes of target people groups (*TPG*) associated with different strategies.

## 2. Literature Review

Recent works [10,11,12] for the spread of COVID-19 mainly include SEIRD, SIRD-RM, and SEIRDV models, and there have been studies on the spread of novel coronavirus pneumonia using these models. A standard model of disease spread is the SEIRD model, in which each individual is either susceptible (*S*), exposed (*E*), infected (*I*), recovered (*R*), or dead (*D*). Compartmental SEIRD models consider only the aggregate number of individuals with each disease state and specify a set of differential equations that govern how the compartmental populations change with time [10].

Reference [11] proposed a compartmental SIRD model with time-dependent parameters that can be used to give epidemiological interpretations to the phenomenological parameters of the Richards growth model. It illustrates the use of the map between these two models by fitting the fatality curves of the COVID-19 epidemic data in Italy, Germany, Sweden, the Netherlands, Cuba, and Japan. The results presented here are relevant in that they showcase the fact that phenomenological growth models, such as the Richards model, are valid epidemiological models not only because they can successfully describe the empirical data but also because they capture, in an effective way, the underlying dynamics of an infectious disease. In this sense, the free parameters of growth models acquire a biological meaning to the extent that they can be put in correspondence (albeit not a simple one) with parameters of compartmental model, which have a more direct epidemiological interpretation.

Reference [12] subdivides the population into six compartments and extends the SEIRD model by adding the vaccinated population and framing the global model as a hybrid-switched dynamical system. Aiming to represent the quantities that characterize the epidemic behavior from an accurate fit to the observed data, they partition the observation time interval into sub-intervals. The model parameters change according to a switching rule depending on the data behavior and the infection rate continuity condition. In particular, they study the representation of the infection rate both as linear and exponential piecewise continuous functions. The authors choose the length of sub-intervals balancing the data fit with the model complexity through the Bayesian information criterion. They tested the model on Italian data and on local data from Emilia-Romagna region. The calibration of the model shows an excellent representation of the epidemic behavior in both cases. Thirty days forecasts have proven to well-reproduce the infection spread though in a way that is better for regional than for national data. Both models produce accurate predictions of infection, but the exponential-based one perform better in most of the cases. Different possible forecast scenarios are obtained by simulating an increased vaccination rate.

They often lack interpretability and behavioral guarantees provided by disease models using differential equations to govern transitions between disease states. However, most of the literature has not considered the impact of super-spreaders on the epidemic and only divided the population into four states, namely susceptible (susceptible, *S*), latent (exposed, *E*), infected (infected, *I*) and removed (removed, *R*), and the classification of lurkers is not comprehensive, such as incubation population in the state of medical observation or suspected cases or latent cases mixed with healthy people (i.e., asymptomatic communicators).

Compared with other literature as shown in Table 1, our main contribution is to provide carefully calibrated and estimated model for assessing the changes of APT and total (including symptomatic infections and asymptomatic patients) of multi-step imperfect screening testing in conjunction with diagnostic testing. By combining a citywide COVID testing model with a nine-group behavioral SEAIRD model, we can consider group-based strategies and the effect of CCT on the development of the epidemic. The model for this strand of the epidemic literature is to enrich the underlying SEIR by introducing a scope for testing policies that may mitigate the output costs of quarantine policies while not exacerbating the decline in output. It would be relatively straightforward to integrate the information structure of our model into an improved SEAIRD model in order to evaluate the available scheme of broad-based testing.

## 3. Epidemiological Model Proposal

### 3.1. Multiple Groups Compartment Model

Larger-scale-screening can be carried out in communities where new outbreaks are emerging in HK. Obviously, no group of people is totally isolated in urban societies. Thus, it is necessary to regard each group of people as reachable nodes. The flow between groups can be described by Markov mobility [13]. In Figure 1, nodes define the different populations in the spread of the disease, and arrows indicate the conversion probability between populations. The model can accurately reflect the spread and outbreak. The model is an infectious disease model with an incubation period [14]. It is designed to calculate the changes in the target population that needs to be screened. It introduces quarantined populations and more groups of people and adds more decision-making factors, which is more complicated but realistic than the prior models.

### 3.2. Parameter Setting

We modeled SARS-CoV-2 transmission and interventions. Because we assume that the community is not a closed population, and there is a risk of continued virus importation, for which we introduce the coefficients of birth and death rates, even in the most optimistic scenario, there is a considerable number of cases mainly from community introductions. Furthermore, this situation requires significant financial and clinical capacity resources. Combined with measures to reduce virus transmission, a testing strategy to identify symptomatic populations, conducting virus testing, and quarantine may be effective in controlling transmission. The success of this strategy relies on contacting tracing and quarantining those with close contact. Screening must be performed at least monthly to have a large impact on the course of the outbreak within the city and greatly increase sample collection and assay requirements.

As shown in Table 2, we divided the populations in the transmission process into compartments of susceptible population (*S*) and exposed population (*E*), infected (symptomatic) (*I*), infected (asymptomatic) (*A*), target groups people (mildly infected population and asymptomatic infected and screened, *TPG*), quarantined and susceptible population (*Sq*), quarantined and exposed population (*Eq*), quarantined and infected population (*Iq*), population that died of a disease (*D*), and recovered population (*R*). First, we assumed contact tracing capabilities and high compliance with quarantine. While changing population size and testing rates is easily achievable, this may be an overly optimistic assumption for some institutions with limited capacity. This value should be parameterized using locally reported incidence data and underreporting estimates. The probability that the susceptible individual is infected and turns into an exposed individual in incubation period (infection rate) is *β*. In particular, a prevalence of 5% was measured in the strongly virus-carrying population, which we expect to be significantly higher than the estimated prevalence in the general population. We assume that infected individuals with strong symptoms have the same infectious rate *φ* as individuals with weak symptoms. The model also takes into account the relative infectiousness of asymptomatic individuals who may have lower infection rates than symptomatic individuals and the probability of conversion from exposed population to asymptomatic population is (1−*ƞ*). There may be significant differences in the prevalence of specific categories of the virus in the general population. In order to facilitate the analysis and calculation, we set the prevalence rate as *θ*. After screening and testing, those diagnosed as positive for COVID-19 were immediately quarantined in the model *ω*. Case quarantine involves completely reducing their contact rate during the infection. Some of these quarantined contacts may have been contagious but are no longer able to infect others once they are in quarantine. The removed patients were differentiated into cured and dead populations, and the conversion rates of the dead population from infected is *γ*, and the conversion rate of cured populations from infected populations is *γ_R_*, which changed with time. In view of the fact that the isolated infected people will be sent to the designated hospital for isolation treatment with a certain probability, according to medical capacity, this part of the population is converted into hospitalized patients in this model. The isolation rate of ρ is mainly affected by the hospital’s medical capacity. Our model is conservative (meaning it may overestimate the COVID burden in cities) because they are more likely to suffer from severe illness and death due to their older age. As the pandemic develops, the immunity of the population is likely to increase, so we added immunization rate to the model. The evolution results of the simulation show that if the population cannot obtain the ability of lifelong immunity, it will be fed back into the model’s circulation system with the system input rate.

In Table 3, we categorize the state parameters into five main parameters with initial accumulation and other parameters without initial accumulation.

### 3.3. Epidemiological Parameters Estimation

For classic and mature models, the reliability of the output results is determined by the accuracy of the input. Therefore, the model starts with specific data and focuses on evaluating, testing, and correcting the input values of the model from multi-angles to improve the model reliability of output. Parameters can be estimated by the method of moments as:(1)$$\hat{\varphi_{t}}=\frac{I_{t+1}-I_{t}}{I_{t}},$$
(2)$$\hat{\varepsilon_{t}}=\frac{I_{t+1}-I_{t}}{E_{t}},$$
(3)$$\hat{\gamma_{R}{}_{t}}=\frac{R_{t+1}-R_{t}}{I_{t}},$$
(4)$$\hat{\gamma_{t}}=\frac{D_{t+1}-D_{t}}{I_{t}},$$
is the number of newly infected patients (*t*~*t* + 1) divided by the number of infectious patients at time *t*. The estimate *εt* is the number of newly infected people (*t*~*t* + 1) divided by the number of exposed people at time *t*. The estimate *γR_t_* is the number of newly recovered people (*t*~*t* + 1) divided by the number of infectious people at time *t*. The estimate *γ_t_* is the number of new deaths (*t*~*t* + 1) divided by the number of infectious people at *t*.

Estimated from the data as of 22 January 2020, Figure 2a shows red scatters as the actual value *φ_t_*, and the red curve is obtained by the method of moments. The estimate at *t* and the confidence interval with the significance of the estimate within 1–0.05 is 0.0304. The trend between the estimate and the actual value is shown in Figure 2b. The largest version of the actual value *ε_t_* and the estimate are at the same level; by calculating the discrete Frechet distance between the actual curves *P* and estimate *Q*, the smaller the value, the more similar the values are. The similarity is 0.0256, and the error between the estimate and true value remains roughly between 0.0007 $\gamma_{R}{}_{t}$ and estimates $\hat{\gamma_{R}{}_{t}}$ are shown in Figure 2c. The largest of the values of the two can be well-controlled on the same level, and the estimate at *t* and the confidence interval of the significance of the estimate of 1–0.05 is 0.0065; the similarity of estimate are calculated as 0.2456. The numerical comparison score between the actual mortality rate and the parameter estimate is 0.061. In Figure 2d, the estimate can fit the actual mortality rate well over time.

Table 4 shows the average estimate of the four parameters obtained. Standard error of the mean (SEM) reflects the degree of fit. The standard error values of the four parameters are small, and the reliability of the simulation prediction is high. Here, the fluctuation of *γ* is small, and the fluctuation of *φ* is relatively large. T-statistic = Ave./Standard_Error. The estimate *γ* is the largest difference from the actual value, and the average value of the time series simulation value has the largest probability of significant difference. The significant differences between the estimated values of *φ* are very small. The *p*-values of the four parameters are all greater than the 5% significance level. There is no significant difference between the experimental group and the real data group.

The trained model is used for simulation with parameter tuning. The HK incidence dataset comes from China’s infectious disease network direct reporting system. In addition, this study was carried out for different regions, and in order to compare with the epidemic data in HK, we chose data from the U.S. with a large difference to further test the robustness and generality of the model. Two independent modeling models were established to monitor trends in various cases. MATLAB was used to simulate the model, and the incidence data from 22 January 2020 to 13 December 2020 were used for model fitting. In Figure 3a, 13 July 2020 and 22 August 2020 are two turning points in the dynamics of the epidemic in HK. The trend of the epidemic from 22 January 2020 to 20 June 2020 and from 7 October 2020 to 13 November 2021 is consistent with the real data. However, the simulated data in the intermediate period (20 June 2020~7 August 2020) are obviously different from the real data, but the overall fit is very good. This difference may be due to the massive social movement that took place in HK on 06/09. In Figure 3b, we show the simulation for the United States because not all infected people can be diagnosed and quarantined immediately. As well as the delay in the spread of the virus, the total accumulative case growth rate was gradual during 28 March 2020~21 March 2020 and 22 March 2020~2 November 2020 and, especially during 2 November 2020~13 November 2020, increased dramatically. In Figure 3c, from 22 January 2020 to 14 July 2020, there were no new deaths, and the cumulative daily deaths from 15 July 2020 to 3 October 2021 also increased relatively slowly. The growth rate of actual cumulative death data from 15 July 2020 to 3 October 2020 was slightly higher than the simulated data. In Figure 3d, for the U.S., the accumulative recovered curve and accumulative death curve both seem to be the same, with an explosive increase in mortality from 30 March 2020 to 27 May 2020. 28 May 2020~13 November 2020 showed steady growth. The error between the fitted data and the expected data points in our model is less than 0.5% of the total number of deaths in the United States. In Figure 3e, in the early stage, no patient was recovered, and the previous confirmed data may be distorted somehow. From 24 March 2020 to 18 July 2020 and from 10 May 2020 to 3 July 2021, there were no excessive increases in accumulative recovered cases. The model can be very close to the actual value. In Figure 3f, as the outbreak dragged on, with intensive contact tracing followed by quarantine interventions, the recovery rate of the recovered population increased on 14 April 2020 and has maintained a steady growth rate. Overall, the model fits the actual data well in the early stage and in the later stage of the epidemic, indicating that the improved model has high prediction precision for COVID-19 and can be used for target people group (TPG) prediction. As already observed, the three models have very similar behavior in the calibration phase, and this behavior is also confirmed by the forecast data reported. Taking the outbreaks of COVID-19 infections in Hong Kong and the United States as examples, we find that the predictions at the inflection point of the epidemic curve have small uncertainties, and the overall situation is basically consistent, with temporal delay in pandemic development and re-susceptibility to temporal immune responses. From Figure 3d,f, it can be found that our model has higher prediction stability, the SEIRDV model has larger prediction volatility, and the SIRD-RM model has the largest prediction error.

In Table 5, according to the parameter estimation method, we obtained four parameter values suitable for the model. Using the simulation experiment of the model, we obtained the value of the natural conversion rate.

The proposed novel model obeys the following rules:Suppose now that a very good performance of the novel coronavirus nucleic acid kit (sensitivity and specificity is 95%) is used to screen all HK people [15].The population is uniformly mixed, and the probability distribution of the positive population is uniformly distributed overall. We can treat it as an absolute uniform distribution.After a patient is cured, he becomes a healthy person who can still be infected again.Models incorporate birth and death.

A mathematical model of infectious diseases was established by using nonlinear dynamics, which is the so-called SETPG (A + I) RD + APT model. According to the system dynamics modeling idea, the following SETPG (A + I) RD + APT dynamic equation with positive and negative feedback is proposed [16]:(5)$$\left\{\begin{array}{c} \begin{array}{c} \frac{\mathrm{dS}\left(t\right)}{\mathrm{dt}}{=v}_{1}\;N\left(t\right)-v_{2}\;S\left(t\right)-\left\{p_{1}\;I\left(t\right)+\varepsilon E\left(t\right)\right\}\left(1-\rho \right)\beta \varphi S\left(t\right) \end{array}\\ -\rho \left(1-\varphi \right)\beta \left\{\varepsilon E\left(t\right)+I\left(t\right)+A\left(t\right)\right\}S\left(t\right){+\omega S}_{q}\left(t\right){+\beta }_{1}E\left(t\right)+\xi R\left(t\right)\\ \frac{\mathrm{dE}\left(t\right)}{\mathrm{dt}}=\left(1-\rho \right)\beta \varphi \left\{\varepsilon E\left(t\right){+p}_{1}E\left(t\right)\right\}-\left({\alpha +\beta }_{1}{+v}_{2}\right)E\left(t\right)\\ \frac{\mathrm{dA}\left(t\right)}{\mathrm{dt}}=\alpha \left(1-\eta \right)E\left(t\right)-\left(\gamma_{R}+\gamma \right)A\left(t\right){-v}_{2}A\left(t\right)\\ \frac{\mathrm{dI}\left(t\right)}{\mathrm{dt}}=\alpha \eta E\left(t\right){+(\gamma }_{R}+\gamma )I(t)\\ \frac{\mathrm{dTPG}\left(t\right)}{\mathrm{dt}}=\mu \left\{A\left(t\right)+I\left(t\right)\right\}\\ \frac{\mathrm{dR}\left(t\right)}{\mathrm{dt}}=\left({1-p}_{1}\right)I\left(t\right)\gamma_{R}-\xi R\left(t\right)\\ \frac{\mathrm{dD}\left(t\right)}{\mathrm{dt}}=\gamma \left\{A\left(t\right)+I\left(t\right)\right\}\\ \frac{{\mathrm{dS}}_{q}\left(t\right)}{\mathrm{dt}}=\rho \left(1-\varphi \right)\beta \left\{\varepsilon E\left(t\right)+I\left(t\right)+A\left(t\right)\right\}S\left(t\right){-\omega S}_{q}\left(t\right)\\ \frac{{\mathrm{dE}}_{q}\left(t\right)}{\mathrm{dt}}=\rho \varphi \beta \left\{\varepsilon E\left(t\right)+I\left(t\right)+A\left(t\right)\right\}S\left(t\right){-\alpha E}_{q}\left(t\right)\\ \frac{{\mathrm{dI}}_{q}\left(t\right)}{\mathrm{dt}}{=\alpha E}_{q}\left(t\right){+\gamma }_{q}\left\{A\left(t\right)+I\left(t\right)\right\}-\gamma_{I}I_{q}\left(t\right) \end{array}\right.$$


In Figure 4, for HK, as of 10 October 2020, the simulated target population is 13,043. The average numbers of asymptomatic and mildly infected are 7174 and 5869, respectively; and for the U.S., the simulated target population is 17,292,960, from which the average numbers of asymptomatic and mildly infected are 9,511,128 and 7,781,832, respectively. A related article pointed out that 30–60% of infections are asymptomatic or having only mild symptoms, but infectivity is not low. We took the average value of all symptomatic infections and the simulated value of asymptomatic to obtain the final average value of the target people group (*TPG*). In Figure 4, the hatched area is the estimated change in the number of cumulative cases of *TPG* in the future obtained by simulation.

## 4. Experimental Simulation

### 4.1. Binary Encoding Nucleic Acid Screening Test

In Table 6, we show the cost saving percentage, time reduction rate, and the positive detection rate as measurement standards to evaluate the pros and cons of different sample pooling results. When the mixing ratio is 5:1~8:1, the cost-saving percentage of sample pooling is 79.5%~86.7%; when the mixing ratio is 10:1~24:1, the cost-saving rate of sample pooling is 89%~93.5%; the time-reduction rate is approximately identical. For positive detection rate, the ratio of different sample pooling is different. The detection rate of 5:1~8:1 is 100%, which is at an ideal state; the detection rate of 10:1 is 83%, which is excellent; the detection rate of 12:1~24:1 is less than half, and the detection rate is low. Under comprehensive consideration, we adopted a screening test with mixing ratio of 10:1.

At different daily detection capabilities (DDC), we used the model to obtain the trend of the target population over time in Figure 5. Grouping detection capabilities help us observe differences in different detection capabilities under different detection groups, thereby selecting the best daily detection volume and providing effective guidance for epidemic prevention.

For HK, with a daily detection capacity of 720,000~750,000, the screening time required for 700,000 and 710,000 are the same, respectively. The former takes up to 10 days and the latter takes 11 days. Generally speaking, the virus has a three-day incubation period. In the middle of the incubation period (about 1.5 days), the virus begins to replicate and become infectious. At this time, the nucleic acid test can detect positive cases. Testing with our decision every other day can improve the scientific accuracy of the mass screening. There is not much difference in time distribution. However, for the number of daily detections, in Table 7, cumulative detection number with a daily detection capacity of 700,000 is 11 days, and the cumulative detection rate is 0.009; and for the U.S., the cumulative detection number with a daily detection capacity of 7,000,000 is 105,360. The cumulative detection rate is 0.015; these two detection effects are the best in HK and the U.S. separately among the group.

At daily detection capacities of 600,000~610,000 and 62,000~660,000, the same screening time is required. The longest time required for a daily detection capacity of 670,000 is 13 days, while the longest duration for DDC of 6,700,000 is 18 days in the U.S. The cumulative detection volume of the target population with a daily detection-capacity of 670,000~660,000 is 6060 and 7404, respectively. The cumulative detection rate of the two are 0.01 and 0.012, for which the required screening time is 12 days and 13 days separately. In the U.S., in the group where the required screening time is 18 days, the cumulative detection numbers of target populations with daily detection capabilities of 6,700,000~6,600,000 are 90,696 and 114,721, and the cumulative detection rates are 0.015 and 0.019, respectively. The detection effect under these four daily detection ability values is the best in these two regions.

At a daily detection capacity of 500,000~580,000, the screening times required for 580,000~560,000, 550,000~520,000, and 510,000~500,000 are about the same, respectively, and cumulative numbers of detections of the target population with a daily detection capacity of 520,000~500,000 are 6150, 7280, and 8153. The cumulative detection rates are 0.012, 0.014, and 0.016, respectively. As for the U.S., the cumulative numbers of target population detections with a daily cumulative detection capacity of 5,200,000~5,200,000 are 102,399, 124,444, and 143,688, respectively, and cumulative detection rates are 0.020, 0.024, and 0.020, respectively. These six daily detection abilities are the best. In contrast, the target detection rate of our model increased by 41.67% in larger-scale screening, which may reflect higher infection and death rates.

At daily detection capacities of 400,000~480,000, 480,000~460,000, 450,000~430,000, and 420,000~410,000, the required screening time is about the same, respectively. The screening time required for a daily detection capacity of 400,000 is 19 days, which is the longest. Cumulative detection numbers of target populations with daily detection capabilities ranging from 420,000 to 400,000 are 7407, 8188, and 9182, respectively. The cumulative detection rates are 0.018, 0.020, and 0.023, respectively. In the U.S., the cumulative numbers of target population detections with daily detection capabilities of 4,200,000~4,000,000 are 144,852, 159,719, and 176,043, and the cumulative detection rates are 0.038, 0.0039, and 0.044. The effect of the six daily detection abilities is the best. We can find that the total screening rate in the United States was 52.63% higher than that in Hong Kong. At a lower daily detection rate, the larger-scale screening rate increased compared with the smaller-scale screening. This may be due to the density of population testing or the accumulation of infections due to lower testing efforts.

Another important parameter that is closely related to infectivity is the effective regeneration number, which measures the ability to spread the disease [17]. It can be simply interpreted as the average number of people infected by an infected person during the infectious period [18]. We made a horizontal comparison of *R0* under each daily detection ability of same groups and obtained the optimal daily detection ability of different groups [19].

In Figure 6, we obtain the *R0* of different daily detection capabilities over time in different groups. In each group, as daily detection capability increases, the minimum timespan (TS) for each virus transmission capability gradually decreases to 0.5 non-linearly, and the rate of decrease gradually accelerates until it stabilizes. In Figure 6a,c,e,g, the shortest timespan (TS) for the virus transmission ability to drop below 1 is 6.5 days, 6 days, 6 days, and 6 days. Likewise, in Figure 6b,d,f,h, the shortest timespan (TS) for the virus transmission ability to drop below 1 is 2.5 days, 1 day, 1 day, and 2 days. With different screening capabilities, we selected two strategies that end the epidemic in the highest DDC/TS rate as the optimal strategy and the average optimization strategy, respectively, for which the TS is 8 with the DDC of 8153, and the TS is 8.5 with the DDC of 9182. For the U.S., the TS of optimal CCT strategy is 9.5 days and 9 days with DDC of 159,719 and 143,688. It can be found that the timespan of citywide COVID testing is basically 1 to 3 days, 5 to 6 days, or 7 to 9 days, which basically conforms to the current policies of two inspections in three days. It also shows the feasibility of our model.

In Table 8, two optimal strategies are obtained, and the results of the average most effective screening strategy are calculated. The screening timespan of the first optimized screening strategy is 8 days in HK and 9.5 days in the U.S. The daily detection capacity of this strategy is 670,000/day in HK and 6,100,000/day in the U.S., and the cumulative detection number is 8153 cases in HK and 2,064,970 cases in the U.S. The total detection timespan of the second optimized screening strategy is 9.5 days in HK and 9 days in the U.S., and daily detection capacity is 580,000/day in HK and 6,000,000/day in the U.S.; and the cumulative detection number is 9182 cases in HK and 1,541,830 cases in the U.S. The detection rates of two strategies are roughly equal: approximately equal to 0.0867 in HK and 1.8034 in the U.S. By averaging the two optimization strategies, we obtain the optimal screening strategy, and total detection timespan is 8.25 days in HK and 9.25 days in the U.S., the total cumulative number of detections (TD) in the target population is 8667 cases in HK and 1,803,400 cases in the U.S., and the detection rate is 0.0867 per 100,000 people in HK and 1.8034 per 1,000,000 people in the U.S., which is about 72 times higher than the current detection rate.

### 4.2. Invisible Virus Catcher—Artificial Intelligence Digital Technology to Achieve Precise Prevention and Control of the Epidemic

In Figure 7, BLE (Bluetooth low-energy) broadcasting technology is considered for use. This technology allows the device to broadcast information to other Bluetooth devices nearby through the Bluetooth low-energy protocol. Since the smart-phone basically has Bluetooth function, this technical solution has designed a way to collect information when contacting other devices at close distance [20]. The app is based on Bluetooth technology. After installation, it will run in the background of the smart-phone and generate an independent anonymous ID. When nearby users with the same app are approaching, it will automatically record the ID through near-field Bluetooth detection. If a user is found to be positive for the virus in the test, the ID will be marked in database. The app cross-checks the lists that match the ID. Once the users in the list are marked as infected, it will send a notification to the users to inform them that they are at risk of exposure. They are then encouraged to contact the local public health department for tests.

When the penetration rate of tracking applications in a country reaches 70%, the accurate tracking of populations can be achieved [21]. In Figure 8, with different app application rates (AAR), the spread rate decreases with the increase of the app application rate. The timespan for the vanishment of the virus transmission capacity is also nonlinearly reduced. When the app application rate is 100% (green curve), that is, in an absolute ideal state, if everyone uses the app to declare their health, the screening timespan required to end the epidemic is 7 days in HK. The cumulative number of people screened within 14 days of the incubation period is 6060. When the app application rate is 90% (light-green curve), that is, in an excellent state, the screening timespan required to end the epidemic is 5.5~6 days. The cumulative number of people screened within 14 days was 2901. When the app application rate is 80% (orange curve), that is, in a good state, the screening timespan required to end the epidemic is 5–5.5 days, and the total number of people screened is 2483. When the app application rate is 70% (red curve), that is, under the conditions of accurate tracking, the screening timespan required to end the epidemic is 5 days, and the cumulative number of people screened is 2065. For the U.S., the AAR is 9, 8, 7, and 5 days separately, corresponding to the total numbers of screening of 114,720, 90,695, 70,540, and 40,376.

### 4.3. Performance

In Table 9, the results through simulation are obtained: in experiment 1, we screened a total of 13,043 cases in HK and 17,292,960 cases in the U.S.; in experiment 2, the total number of samples screened was 8667 cases in expt. 1 and 6060 in expt. 2 for HK and 1,803,400 cases in expt. 1 and 1,766,420 cases in expt. 2 for the U.S., which are 11,702 screening cases relatively reduced compared to experiment 1, improved screening rate by at least 53.54%.

In Table 10, through different performance indicators, we can see that the sensitivity and specificity of experiment 2 are slightly higher than those of experiment 1. The false positive rate of experiment 2 is higher than that of experiment 1, and the false-negative rate is significantly higher than that of experiment 2. The +LR of experiment 2 is higher than that of experiment 1. For −LR performance index, experiment 2 is also lower than experiment 1; that is, the diagnostic value of experiment 2 is higher than that of experiment 1. For accuracy, experiment 2 is significantly higher than experiment 1, which reflects the superiority of experiment 2.

## 5. Discussion

Citywide COVID testing could provide the means for regular and mass screening for the early detection of asymptomatic and symptomatic individuals, an especially important aspect of successful epidemic containment. We present examples of forecasts for viral transmission in the Hong Kong and the United States. This work has demonstrated how to build a novel model for the COVID-19 outbreak and screening in different regions, including interventions, estimating model parameters, and generating posterior predictive intervals. Furthermore, the model is able to treat the asymptomatic and mild symptomatic compartments as target variables, as no detailed data were observed about them other than approximate initial values. For diagnostic accuracy, the model fits the data quite well with an AC ≈ 60.11 for HK and AC ≈ 95.02 for the U.S. in experiment 1 and predicts reasonably well with an AC ≈ 62.70 for HK and AC ≈ 95.21 for the U.S. in experiment 2. One can also note that in the model definition, quarantine, vaccination, natural births, and natural mortality were included.

Major hospitals are faced with screening and diagnosis tension during the pandemic. To carry out screening in a short time, screening tests in different models were carried out. No system is perfect, and the accuracy is relatively low. There is always some uncertainty and risk, which result in false positives or false negatives. All-people testing combined with the virus-tracking system is more specific than the nucleic acid test alone. The false negatives of the two screening methods are much lower than the false-positive rate, which is inevitable. The false-positive rate of nucleic acid testing is 1.4 times higher than the false-positive rate of the virus-tracking system. Compared with the false-negative rate, the false-positive rate is significantly higher, which can easily cause the collapse of medical treatment. Combining false-positive and false-negative results, if the specificity of a disease test is high (the research indicates that it is about 50.03% in expt. 1 and 66.26% in expt. 2 for HK and 66.26% in expt. 1 and 66.24% in expt. 2 for the U.S.), and the diagnostic result is positive, the person may have the disease and decide whether to conduct other tests. Tests with high specificity are mostly used for deriving rule in a diagnosis because there is less chance for being ignored. The two diagnostic tests have the ability to distinguish a case from a certain disease. The greater the +LR of expt. 1 (1.4104 for HK and 2.9520 for the U.S.) and 2 (1.9841 for HK and 2.9568 for the U.S.), the greater the probability of achieving a true positive when the test result is positive. In –LR, the smaller the ratio of expt. 1 (0.8900 for HK and 0.0142 for the U.S.) and 2 (0.4368 for HK and 0.0027 for the U.S.), the greater the probability that the result will be truly negative.

Future work could be to include more parameters (e.g., consider time-dependent infection rates or viral load in patients after infection or graphs within communities) in the model to capture uncertainty more accurately. Furthermore, one could be employed to perform simulation studies in order to obtain conclusive results that can be used as healthcare guidelines and better elucidate how the model may perform under various scenarios. Feature selection methods can be used to select where the policy interventions should be placed, and other forms of interventions could be included in the model. Another possibility to address any deviations from the standard model is a semiparametric technique, which could be studied as well.

## 6. Conclusions

The data collected from HK were applied to a novel infectious disease model called SETPG (A + I) RD + APT, which enabled a simulation experiment of the all-people test and compared results with U.S. data to validate the model robustness. By combining coding nucleic acid detection and app virus-tracking system, the applicability of the all-people test was improved. The infectious disease model established has good prediction accuracy and can be used for short-term predictions for universal screening. When dealing with the all-people test, this research encountered the problems of overloaded testing, slow speed, and long training time. In order to solve this problem, this study used the two experimental methods of coding method and tracking system to optimize the simulation model, thereby obtaining an optimized strategy for a short-term all-people test suitable for the reality. In order to improve the performance and make the model closer to reality, we added different populations and prevention measures to the model. Using mathematical methods to analyze the mathematical model based on an all-people test of 10 populations, the physiological significance, prevention, and control mechanisms were obtained. Finally, multi-evaluation factors in the screening evaluation were used to comprehensively show the performance results of the two experiments. Results show that both experiments can improve the efficiency and effectiveness of the all-people test. Here, the advantage of using an app virus-tracking method is more prominent, and the result of the experiment is also in line with reality.

This model, however, has limitations. The data for establishing the SETPG (A + I) RD + APT model are a dynamic time series. Data collection is affected by factors such as timeliness and control measures. The model will inevitably be different from reality, which will lead to a deviation in analysis and prediction results of the target people group. The parameters of the model are estimated through the literature and mathematical model fitting, and there may be a deviation between the predicted results and actual results. Observing and predicting from the data, it is relatively simple and easy to implement, and the short-term prediction effect is acceptable. However, the long-term prediction performance is significantly reduced, and only variable prediction results can be given, which cannot reveal the deeper law of the development of the epidemic.

## Figures and Tables

**Figure 1 ijerph-19-10959-f001:**
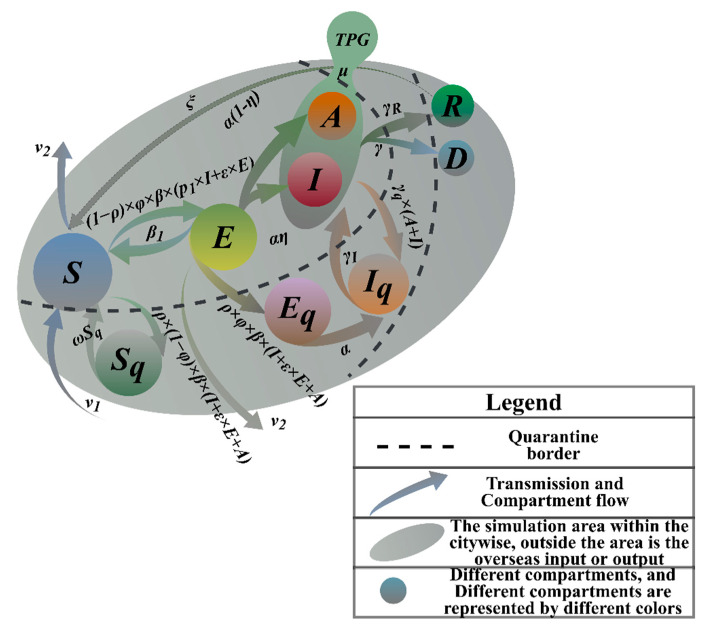
SETPG (A + I) RD + APT Imitation Dynamics Model Frame Diagram.

**Figure 2 ijerph-19-10959-f002:**
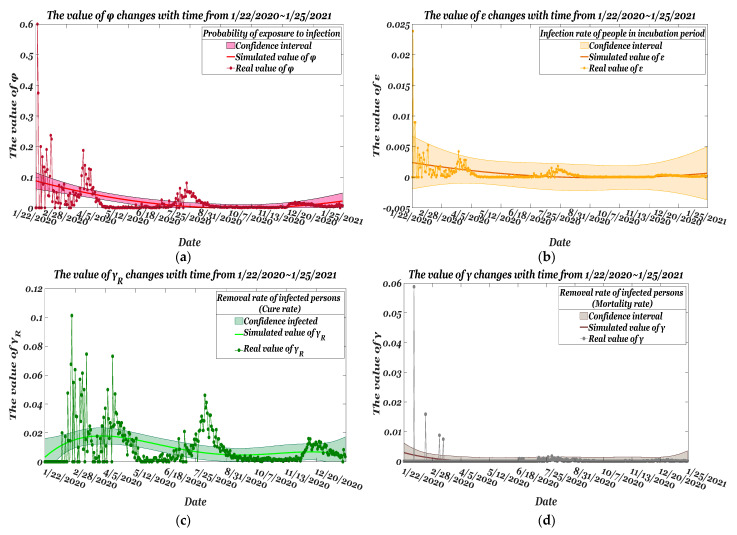
Comparison of real parameters and the varying rate of estimated parameters from 22 January 2020~25 January 2021: (**a**) *φ*; (**b**) *ε*; (**c**) *γ_R_*; (**d**) *γ*.

**Figure 3 ijerph-19-10959-f003:**
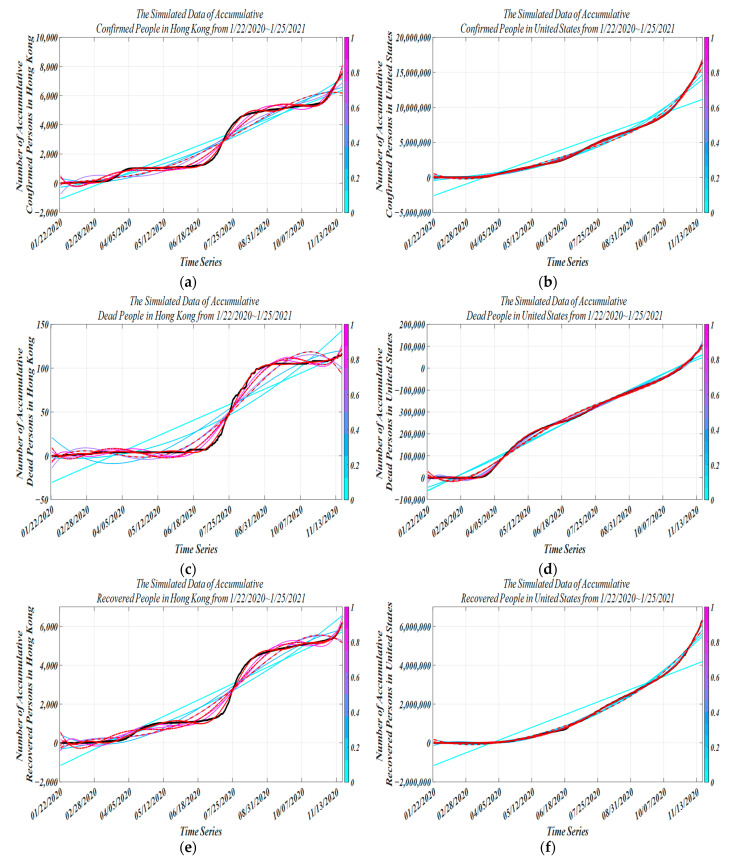
Simulated data vs. actual data from 22 January 2020 to 25 January 2021 (black curve, actual data of accumulative dead people; other curves, simulated data of accumulative dead people; different color curves, different degrees of fitness; red solid line, our methodology; red dashed line, SIRD-RM model; red dashed line with dots, SEIRDV model) (**a**) vs. (**b**) accumulative confirmed people (**c**) vs. (**d**) accumulative recovered people (**e**) vs. (**f**) accumulative dead people.

**Figure 4 ijerph-19-10959-f004:**
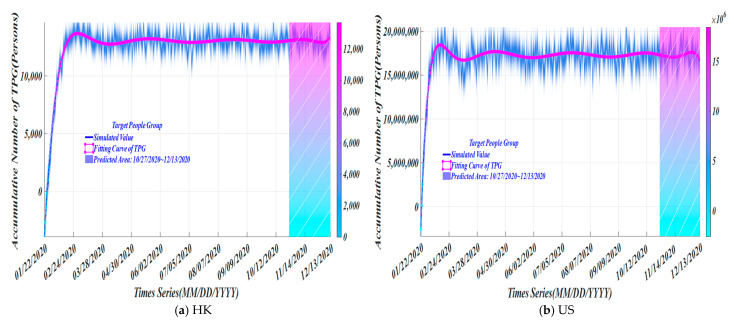
The trend of simulated number of *TPG*.

**Figure 5 ijerph-19-10959-f005:**
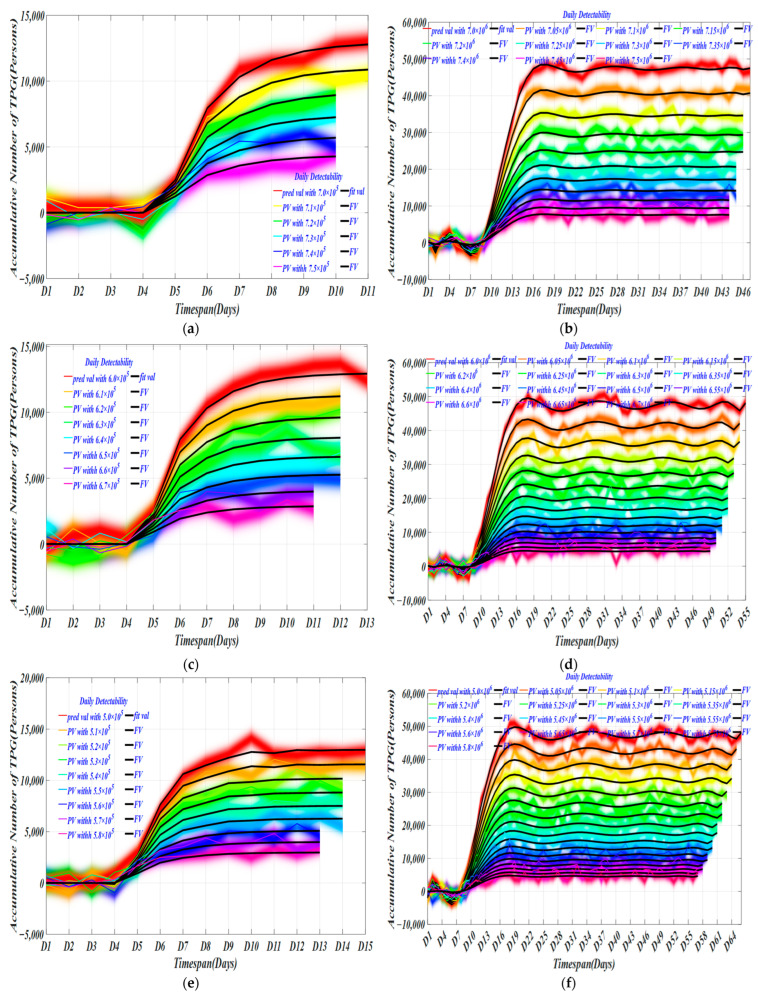
Changes in the number of TPG with different daily detectability. (**a**) The daily detectability from 700,000 cases to 750,000 cases with an interval of 10,000 cases in HK; (**b**) The daily detectability between [7,000,000 cases 7,500,000 cases] with an interval of 50,000 in US; (**c**) The daily detectability between [600,000 cases 670,000 cases] with an interval of 10,000 in HK; (**d**) The daily detectability between [6,000,000 cases 6,700,000 cases] with an interval of 50,000 in US; (**e**) The daily detectability between [500,000 cases 580,000 cases] with an interval of 10,000 in HK; (**f**) The daily detectability between [5,000,000 cases 5,800,000 cases] with an interval of 50,000 in US; (**g**) The daily detectability between [400,000 cases 480,000 cases] with an interval of 10,000 in HK; (**h**) The daily detectability between [4,000,000 cases 4,800,000 cases] with an interval of 50,000 in US.

**Figure 6 ijerph-19-10959-f006:**
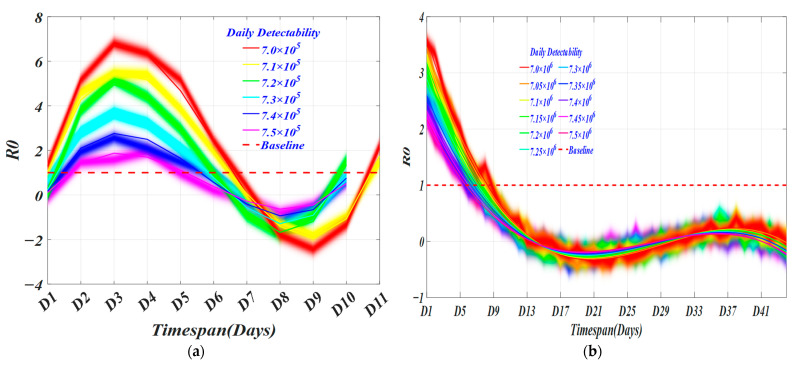
The *R0* of different daily detectability changes over time. (**a**) The daily detectability from 700,000 cases to 750,000 cases with an interval of 10,000 cases in HK; (**b**) The daily detectability between [7,000,000 cases 7,500,000 cases] with an interval of 50,000 in US; (**c**) The daily detectability between [600,000 cases 670,000 cases] with an interval of 10,000 in HK; (**d**) The daily detectability between [6,000,000 cases 6,700,000 cases] with an interval of 50,000 in US; (**e**) The daily detectability between [500,000 cases 580,000 cases] with an interval of 10,000 in HK; (**f**) The daily detectability between [5,000,000 cases 5,800,000 cases] with an interval of 50,000 in US; (**g**) The daily detectability between [400,000 cases 480,000 cases] with an interval of 10,000 in HK; (**h**) The daily detectability between [4,000,000 cases 4,800,000 cases] with an interval of 50,000 in US.

**Figure 7 ijerph-19-10959-f007:**
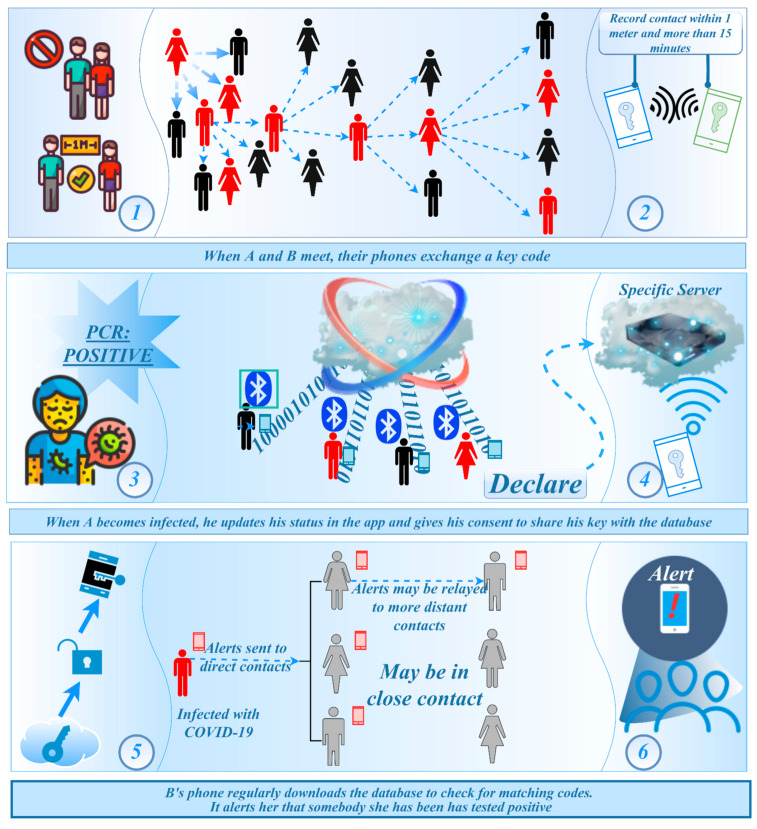
Sketch-map of Bluetooth wireless technology virus-tracking mechanism. (Note: Step 1: effective contact distance for collecting close contacts’ information; Step 2: When close contact conditions are reached, the mobile device starts recording information; Step 3: Someone get a positive PCR test result for COVID-19 (coronavirus) and confirmed cases were screened; Step 4: Through the cloud data, the records of the confirmer’s equipment are retrieved to obtain the people who have had close contact behaviors; Step 5: The background system will automatically pair through the key of the confirmed patient, and obtain the bluetooth signal key of all the user’s mobile phones that they have been in close contact with within a period of time; Step 6: Once a confirmed case appears, the mobile phone that has been in close contact with the patient’s mobile phone will also automatically warn its owner, reminding its owner that the person you have been in contact with has been diagnosed and needs self-quarantine and seek medical treatment as soon as possible.)

**Figure 8 ijerph-19-10959-f008:**
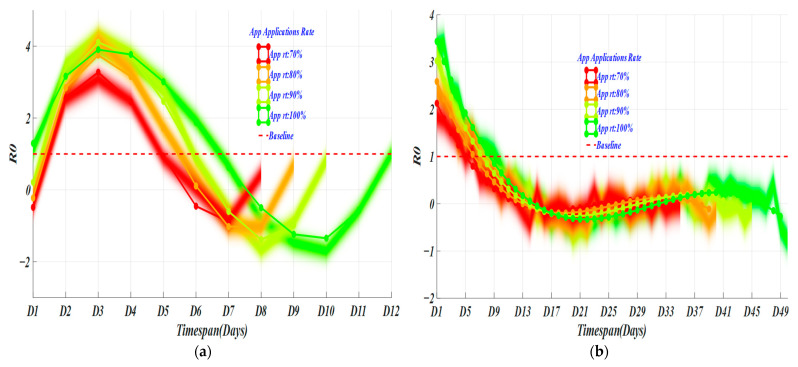
Epidemic development trends of different app application rates. (**a**) the changes of *R0* in Hong Kong; (**b**) the changes of *R0* in United States.

**Table 1 ijerph-19-10959-t001:** Research work found in the literature for disease prediction.

Method (s)/Type of Modeling	Pros	Cons	The Research Found in the Literature
SEIRD model	The model has good prediction ability and decent performance. Obtained long-term predictions reflect the general dynamic of the outbreak and are especially useful for the healthcare system workers and government officials. Obtained short-term predictions allow us not only to forecast the future number of infected, recovered, and deceased patients but also estimate forecast error under adverse or optimistic circumstances. The proposed method can be used as an effective tool for prediction and analysis of the dynamics of the COVID-19 pandemic.	The model does not consider that the exposed category may have a partial infection ability and does not distinguish symptomatic from asymptomatic people.	Maher et al., 2021 [10]
SIRD-RM growth models	The Richards models are valid epidemiological models not only because they can successfully describe the empirical data but also because they capture, in an effective way, the underlying dynamics of an infectious disease. In this sense, the free parameters of growth models acquire a biological meaning to the extent that they can be put in correspondence with parameters of compartmental model, which have a more direct epidemiological interpretation.	Due to lack of knowledge of the epidemiological cycle and absence of any type of vaccine or medications, the government issued various non-pharmacological measures to end the COVID-19 pandemic.	Macêdo et al., 2021 [11]
Switched forced SEIRDV compartmental models	The model introduces a model extension that takes into account the reduced vaccine efficacy and presents a preliminary experiment in the hypothesis of mass vaccination with a single vaccine dose.	The possibility of reinfection and the difference between one-dose and two-dose vaccinations were not considered.	Erminia et al., 2022 [12]

**Table 2 ijerph-19-10959-t002:** Parameter Definition and Meaning.

State Para			
Symbol		Meaning	
Non-quarantined state	*S*	Susceptible people	
	*E*	In incubation period	
	*A*	Asymptomatic patients	
	*I*	Infected population	
Quarantine state	*TPG*	The target population (including asymptomatic infections and confirmed populations with mild symptoms)	
	*Sq* ^1^	Isolated susceptible people	
	*Eq* ^1^	Isolated people in incubation period, no risk of infection	
	*Iq* ^2^	Isolated infections without risk of infection	
Others	*R*	Cure (fully recovered, not fully recovered)	
	*D*	Dead	
**Conversion Para**			
**Symbol**		**Description**	**Meaning**
*φ*		Infection rate	Probability of exposure to infection
*ε*		Relative infection ratio	Infection rate of people in incubation period
*α*		Positive feedback	Probability that people in incubation period will turn positive
*ƞ*		Input rate	Ratio of symptomatic infections to all infections
*β*		Exposure rate	Rate of exposure of susceptible people to people in incubation period or the infection
*β_1_*		Immunization rate	Probability that people in incubation period will turn negative
*ξ*		System input rate	The ratio of reverting to susceptibility after recovery
*p_1_*		Input rate	Ratio of mild infections to symptomatic infections
*μ*		Daily detection rate	Daily nucleic acid testing capacity
*θ*		Prevalence rate	The proportion of people suffering from a certain disease at a certain period of time
**Removal Rate**			
**Symbol**		**Description**	**Meaning**
*γ*		Mortality rate	Removal rate of infected persons (mortality rate)
*γ_R_*		Cure rate	Removal rate of infected persons (cure rate)
**Quarantined Rate**			
**Symbol**		**Description**	**Meaning**
*ω*		De-quarantine rate	Isolation rate of susceptible population
*γ_q_*		Overload rate of medical conditions	Probability of quarantined infected person
*γ_I_*		Treatment rate	Probability of hospital admission from isolation state
ρ		Quarantined rate	The proportion of medical observation subjects in the real epidemic
**Natural Conversion Rate**			
**Symbol**		**Description**	**Meaning**
*ν* _1_		Input rate	Birth rate
*ν* _2_		Output rate	Natural mortality rate

^1^ represents the suspected population; ^2^ represents the confirmed population.

**Table 3 ijerph-19-10959-t003:** The State Parameter of Model.

Parameters Setting Class	State Parameter				
	Term		Simulated Value	Fixed Experience Value	Relative Error (%)
	No.	Parameters			
**With initial accumulation value**	1	*S*	1 × 10^5^	1 × 10^5^	0
	2	*E*	45	1 × 10^3^	0.9550
	3	*A*	0	10	1
	4	*I*	6	20	0.7000
**Without initial accumulation value**	5	*TPG* ^1^	-	-	0
	6	*Sq* ^1^	-	-	0
	7	*Eq* ^1^	-	-	0
	8	*Iq* ^1^	-	-	0
	9	*R* ^1^	-	-	0
	10	*D* ^1^	-	-	0

^1^ Simulated value ^1^ = 0; fixed experience value ^1^ = 0.

**Table 4 ijerph-19-10959-t004:** Error analysis of four main parameters.

Term	Estimate	SEM	t-Statistic	TSTAT ^1^	*p*-Value (pr > |*t*|)
φ	0.02385	0.0037	6.447	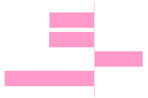	0.05590
ε	5.2844 × 10^−4^	8.1374 × 10^−5^	6.494		0.27123
$\gamma_{R}$	0.0095	7.0950 × 10^−4^	13.395		0.13234
γ	3.4812 × 10^−4^	1.6785 × 10^−4^	2.074		0.05877

^1^ Here we take 6.447 as the baseline.

**Table 5 ijerph-19-10959-t005:** Conversion Parameter Settings.

Class		Value	Basis
**Parameter estimate**	φ	0.1198	Model simulation + Method of moments
	ε	0.000986415	
	$\gamma_{R}$	0.107358	
	γ	0.0063073	
**Conversion rate**	α	0.000187	Model simulation
	ƞ	0.001	
	β	0.0009	
	β1	0.1	
**Quarantine rate**	ω	0.0714	1/Quarantine days
	γq	0.1	1/Course of disease
	γI	0.1	
**Natural conversion rate**	ν1	0.1	Model simulation

**Table 6 ijerph-19-10959-t006:** Performance Results of Nucleic Acid Detection with Different Mixing Ratios.

Group (Accepted ✔ or Non-Accepted ×)	Group Testing *k*	Average Number of Screenings (Person)	Time Reduction Rate (%)	Single Tube Detection Capability (before Mixing Experiment)	Solution Concentration (mL)	Positive Detection Rate (%)	Application Area
A✔	5:1	0.2050	7.1428	50,000 tubes/day (the daily testing volume of private hospitals and private laboratories)~100,000 tubes/day (Huo-Yan air membrane laboratory)	3000	100	Beijing, Xinjiang
B✔	6:1	0.1727	-		2000	100	-
C✔	8:1	0.1330	6~7		1000	100	Shanghai
D✔	10:1	0.1100	6.5175		500	83	Hubei, Wuhan
E×	12:1	0.0953	-		250	50	-
F×	24:1	0.0654	-		100	17	-

Source: Wuhan Municipal Health Commission Daily News.

**Table 7 ijerph-19-10959-t007:** Model Simulation Results of the All-People Test Based on Binary Coding HK/US.

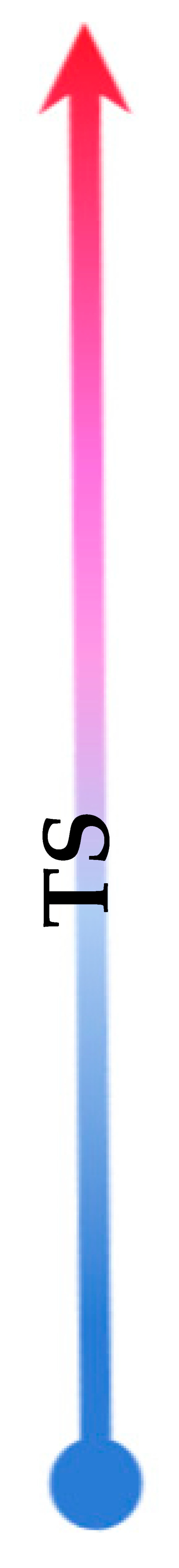	**TS**	**TD**	**DR**	**TS**	**TD**	**DR**	**TS**	**TD**	**DR**	**TS**	**TD**	**DR**	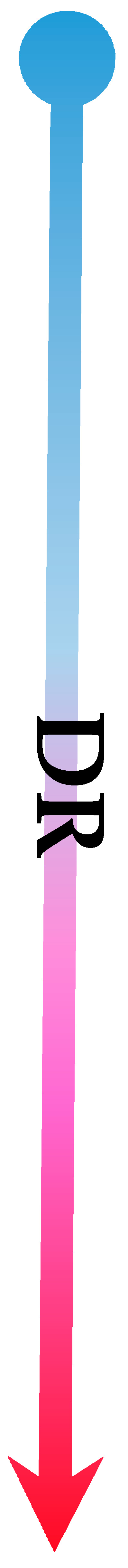
11/16	63,939/105,360	0.009/0.015	13/18	7404/114,721	0.012/0.019	15/18	8153/143,688	0.016/0.020	19/19	9182/176,043	0.023/0.044
11/15	5459/79,803	0.008/0.011	12/17	6060/90,696	0.01/0.015	15/16	7280/124,444	0.014/0.024	18/17	8188/159,719	0.02/0.039
10/15	4118/56,976	0.006/0.008	12/16	5211/70,540	0.008/0.011	14/15	6150/102,399	0.012/0.020	18/15	7407/144,852	0.018/0.034
10/14	3368/41,251	0.005/0.006	12/15	4398/53,893	0.007/0.009	14/14	5350/83,141	0.01/0.016	17/13	6454/131,711	0.015/0.031
10/13	2670/29,139	0.004/0.004	12/14	3626/40,376	0.006/0.006	14/13	4575/66,545	0.008/0.012	17/12	5706/112,371	0.013/0.026
10/13	2027/19,065	0.003/0.003	12/13	2901/29,601	0.004/0.005	14/12	3831/52,450	0.007/0.010	17/10	4971/102,735	0.011/0.023
	11/13	2065/20,075	0.003/0.003	13/11	2971/40,662	0.005/0.007	16/8	4128/95,971	0.009/0.021
11/12	1501/13,897	0.002/0.002	13/10	2342/30,968	0.004/0.005	16/7	3460/81,653	0.007/0.017
	13/9	1761/23,137	0.003/0.004	16/5	2821/81,792	0.006/0.017
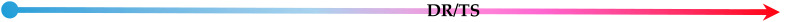

**Note:** TS is short for timespan; DR is short for detection rate. The color of the energy bar changes with the value, the lower the value, the darker the blue, and the higher the value, the darker the red.

**Table 8 ijerph-19-10959-t008:** Model Simulation Optimization Results Combined with Coding Screening Method.

TS (Days)	Region	Total Detection Number (Persons)	Asymptomatic Infection Detected (Persons)	Mild Infection Detected (Persons) ^1^	Detection Rate (Per 100,000 and 1,000,000 People)	Residual Rate of Total Infected Persons
8/9.5	HK/U.S.	8153/2,064,970	5708/1,445,479	2445/619,491	0.0815/2.0650	−0.0547
8.5/9	HK/U.S.	9182/1,541,830	6427/1,079,281	2754/462,549	0.0918/1.5418	−0.0413
**Average Screening Result**						
8.25/9.25	HK/U.S.	8667/1,803,400	6067/1,262,380	2599/541,020	0.0867/1.8034	−0.048

^1^ Mild infection detection: calculated based on the proportion of people with mild symptoms of 0.3.

**Table 9 ijerph-19-10959-t009:** Confusion matrix of two experiments.

Screening Experiment 1			Actual Data/Gold Standard		Total
			P(+) PatientHK/U.S.	N(−) Non-Patient HK/U.S.	
Encoding All-People Test	Simulated Data	P(+)	TP = 7541/16,408,710	FP = 5502/884,250	13,043/17,292,960
		N(−)	FN ^1^ = 3159/66,581	TN = 5508/1,736,819	8667/1,803,400
	Total		10,700/16,475,291	11,010/2,621,069	21,710/19,096,360 ^2^
**Screening** **Experiment 2**			**Actual Data/Gold Standard**		**Total**
			**P(+) HK/U.S.**	**N(−) HK/U.S.**	
App Epidemic-Tracking Platform	Simulated Data	P(+)	TP = 7541/16,408,710	FP = 5502/884,250	13,043/17,292,960
		N(−)	FN ^1^ = 2262/29,601	TN = 5508/1,736,819	6060/1,766,420
	Total		10,442/16,438,311	8661/2,621,069	19,103/19,059,380 ^2^

^1^ FN, count (AC + mild symptomatic patients); ^2^ Total is the simulated results in predicted model; T stands for test, and P stands for patient. TP, true positive; FP, false positive (Type Ι error); FN, false negative (Type ΙΙ error); TN, true negative; (for comparison with below: T+P+ = TP, T−P− = TN, T+P− = FP, and T−P+ = FN).

**Table 10 ijerph-19-10959-t010:** Evaluation Results of Screening Plan for Comparison.

Evaluation Index	Formula (×100%)	Experiment 1% HK/U.S.	Experiment 2% HK/U.S.
**Sensitivity, sen**	ΣTP/Σ (TP + FN)	70.48/99.60	72.22/99.82
**Specificity, sp**	ΣTN/Σ (TN + FP)	50.03/66.26	63.60/66.24
**False-positive rate, Fpr**	ΣFP/Σ (FP + TN)	49.97/33.74	63.53/33.74
**False-negative rate, Fnr**	ΣFN/Σ (TP + FN)	29.52/0.40	21.66/0.18
**Positive likelihood ratio, +LR**	sen/(1 − sp)	1.4104/2.9520	1.9841/2.9568
**Negative likelihood ratio, −LR**	(1 − sen)/sp	0.8900/0.0142	0.4368/0.0027
**Accuracy, AC**	Σ(TP + TN)/Σ(TP + TN + FP + FN)	60.11/95.02	62.70/95.21

## Data Availability

The data that support the findings of this study are openly available in Kaggle at https://www.kaggle.com/imdevskp/corona-virus-report (accessed on 5 July 2021).

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
