# Peer review of "All-People-Test-Based Methods for COVID-19 Infectious Disease Dynamics Simulation Model: Towards Citywide COVID Testing"

_ijerph, 2022, doi:10.3390/ijerph191710959_

Round 1
Reviewer 1 Report
After the revision of this manuscript, the overall framework and the quality of the article have been significantly improved. This research topic has maintained innovation while closely combining with practice. The following changes are worth mentioning.
1. In the revised manuscript, it is noted that the author has added highlights and problem statements in the Introduction, which plays a good role in emphasizing the significance and contribution of this paper.
2. The author added the Literature Review part to make the overall structure of the article more complete. It is worth mentioning that the author added a comparison table of related work in the Literature Review chapter, which well shows the relationship between solution and past literature.
3. The quality of statistical analysis pictures in the revised manuscript has been significantly improved, which is very attractive. At the same time, it also helps to clearly show the experimental results.
4. In the two chapters of Epidemiological Model Proposal and Parameter Setting, the previous situation of piling up too many references for a concept description has changed.
Reviewer 2 Report
Authors updated the paper as per my previous comments. No update requires from my side.
This manuscript is a resubmission of an earlier submission. The following is a list of the peer review reports and author responses from that submission.
Round 1
Reviewer 1 Report
This research has developed a practical dynamic infectious disease model-SETPG(A+I)RD+APT, which fits all epidemic cases and improves the applicability of all-people-test. The research topics closely follow the reality, and the work has a certain degree of innovation. However, I want to focus on a few things, and I hope the author can improve the article for the following points.
- The overall structure of the article is very full, but I think the Literature Reviewpart is missing. It is recommended that the author separate the relevant content from the Introduction chapter to supplement the Literature Review chapter to illustrate the inspiration and innovation of past research on this
- In the two chapters of Epidemiological Model Proposaland Parameter Setting, for the description of a concept or a fact, the citations of the literature should not be piled up too much. Similar situations in this article seem to appear a bit, which will affect the quality of the article. I hope the author will make some adjustments in the future.
- In the second chapter, the selection of the infectious disease model did not show some good reasons for choosing the model. It is recommended to add some of the author's own thought contributions to the advantages of the model and related literature materials here.
- In the Experimental Simulation chapter, the statement of the experimental results is too long. It is recommended that the author can appropriately add some of his own analysis and opinions on the data, and add some practical significance of the modelto highlight the innovation and scientificity of the article.
- There are deviations between the title of some chapters and subtitles in this article. I hope the author will check it carefully to avoid reading obstacles.
Authors' Reply to Reviewer 1
|
Q1. The overall structure of the article is very full, but I think the Literature Review part is missing. It is recommended that the author separate the relevant content from the Introduction chapter to supplement the Literature Review chapter to illustrate the inspiration and innovation of past research on this |
|
A1. A Literature Review part has been added; see p. 4~5 that I highlighted. |
|
Q2. In the two chapters of Epidemiological Model Proposal and Parameter Setting, for the description of a concept or a fact, the citations of the literature should not be piled up too much. Similar situations in this article seem to appear a bit, which will affect the quality of the article. I hope the author will make some adjustments in the future. |
|
A2. Thank you for your suggestions, we made changes to address the reviewer’s comment. |
|
Q3. In the second chapter, the selection of the infectious disease model did not show some good reasons for choosing the model. It is recommended to add some of the author's own thought contributions to the advantages of the model and related literature materials here. |
|
A3. we agree about the lack of clarity. We added the detailed as recommended and have ensured that the information is now included in the text. |
|
Q4. In the Experimental Simulation chapter, the statement of the experimental results is too long. It is recommended that the author can appropriately add some of his own analysis and opinions on the data and add some practical significance of the model to highlight the innovation and scientificity of the article. |
|
A4. In the present paper, we have modified and added some explanations to clarify it. |
|
Q5. There are deviations between the title of some chapters and subtitles in this article. I hope the author will check it carefully to avoid reading obstacles. |
|
A5. Thanks for your kind reminders. We have made revisions accordingly. |
Reviewer 2 Report
The article evaluates a new and interesting subject, but some aspects need to be clarified. The article fails to establish the state of the art in the literature and justifying the choice and development of the model.
Other remarks:
What HK means? (line 22)
Abstract: What´s the research gap?
Please, clarify in the Introduction (line 39): What is the “research question” ?.
Are cases quantities integer or decimal numbers? (line 29)
Paragraph starting in line 65: What´s the source for these statements?
Paragraph starting in line 150: What´s the source for these statements?
What is the theoretical basis for the development of the model?
What is the state of the art in literature?
Are there other similar epidemic models available?
Why did the authors develop and choose the proposed model?
Have other models been compared?
Have other locations been compared? Is it possible to apply in other places?
Please, improve the Discussion topic (line 385)
Authors' Reply to Reviewer 2
|
Q1. What HK means? (line 22) |
|
A1. we thank you for raising this issue. We remove abbreviation “HK” and replaced it by “Hong Kong”. |
|
Q2. Abstract: What´s the research gap? |
|
A2. This is now detailed in the abstract which is highlighted. |
|
Q3. Please, clarify in the Introduction (line 39): What is the “research question”? |
|
A3. We clarified the research question in the introduction section which is highlighted. We hope it is clearer. |
|
Q4. Are cases quantities integer or decimal numbers? (line 29) |
|
A4. The advantages of the dynamical equation have been exploited on a large scale by the research community to obtain approximate solutions to many problems that arise in a wide range of applied science and technology. The optimal parameters that can be obtained through iteration, during the period, considering that the number of people that can be obtained by multiplying the directly given ratio by the number of people is not an integer, and the predicted number of people extracted is also a decimal. To avoid misunderstanding, the number of cases obtained in the model will be rounded up. |
|
Q5. Paragraph starting in line 65: What´s the source for these statements? Paragraph starting in line 150: What´s the source for these statements? |
|
A5. We have added relevant literature to explain. Nucleic acid extraction and detection processes are in accordance with the relevant laboratory standard operating procedures, one is to sample several people, such as 5 or 10 people, respectively, in different "mixed sampling" modes, which will not affect the sensitivity of nucleic acid detection. |
|
Q6. What is the theoretical basis for the development of the model? |
|
A6. Developing a proper understanding of the dynamics of the COVID-19 epidemic curves is an ongoing challenge. In modeling epidemics, in general, compartmental models have been to some extent the tool of choice. Based on the theoretical model of the standard model SEIRD, we carry out population refinement and policy intervention of the model to obtain a more realistic and novel theoretical model, which is named SETPG(A+I)RD+APT dynamics model. |
|
Q7. What is the state of the art in literature? |
|
A7. our work is the first to use a citywide covid testing to model the relationship between time, prevalence, and infection rate, the first to use more complex population group networks for the purpose of more realistic epidemic behavior and development, and also the first to fit a single model to multiple epidemic development from across the different scenarios. We have added the literature in the Pg4-5, Ln153-221. |
|
Q8. Are there other similar epidemic models available? Have other models been compared? |
|
A8. There are few similar models that can be directly compared with our methodology for citywide covid testing, for this reason, our research field is still a research gap. However, we used other recent epidemic models to compare the epidemic curve prediction, highlighting the feasibility of our model. |
|
Q9. Why did the authors develop and choose the proposed model? |
|
A9. We know most cases of the coronavirus go undetected, often because people don’t even realize that they are infected. They might have no, mild or unusual symptoms, for instance, but can still infect others. If a bigger proportion of the people carrying the virus can be identified and isolated, there will be a bigger reduction in the spread of the virus. Citywide covid testing could provide the means for regular and mass screening for the early detection of asymptomatic and symptomatic individuals - an especially important aspect of successful epidemic containment. This has been mentioned in the abstract. |
|
Q10. Have other locations been compared? Is it possible to apply in other places? |
|
A10. We have added the different region, US, for the comparison through the experiment. |
|
Q11. Please, improve the Discussion topic (line 385) |
|
A11. we clarify in the discussion section accordingly (lines 539–580). |
Reviewer 3 Report
The article is well written and easy to understand. However, few of my feedback can be considered to improve the quality of the paper but all are not necessary.
- Introduction may be improved, adding the highlights and the problem statements.
- Figure quality is very poor and can be improved.
- Review references because some of them are unstandardized.
- The difference between your proposal and related works is not clear, you could to details better. I suggest add a comparative table in ''Related Literature'' to contrast your solution in front of related works.
- You could discuss the relationship between your solution and past literature.
Authors' Reply to Reviewer 3
|
Q1. Introduction may be improved, adding the highlights and the problem statements. |
|
A1. Following your suggestion, we have attempted to add the highlights and the problem statements part in the Introduction section which it is highlighted. |
|
Q2. Figure quality is very poor and can be improved |
|
A2. Thank you for the nice reminder. We have improved most of the figure to make them clearer. |
|
Q3. references because some of them are unstandardized. |
|
A3. Thank you very much for pointing this out. We have gone through your comments carefully and tried our best to address them one by one. |
|
Q4. The difference between your proposal and related works is not clear, you could to details better. I suggest add a comparative table in ''Related Literature'' to contrast your solution in front of related works. You could discuss the relationship between your solution and past literature. |
|
A4. Thanks for your comment. I have added a paragraph in the literature review in p. 4~5 which it is highlighted. |